# Predicting Favorable Conditions for the Determination of Initial Use of Janus Kinase Inhibitors in Patients with Moderate to Severe Atopic Dermatitis

**DOI:** 10.3390/jcm14124312

**Published:** 2025-06-17

**Authors:** Ju Heon Park, Sejin Oh, Jihye Park, YoungHwan Choi, Jong Hee Lee

**Affiliations:** 1Department of Dermatology, Samsung Medical Center, School of Medicine, Sungkyunkwan University, Seoul 06351, Republic of Korea; juhun98@naver.com (J.H.P.); sejin5474@daum.net (S.O.);; 2Department of Medical Device Management & Research, Samsung Advanced Institute for Health Sciences & Technology (SAIHST), Sungkyunkwan University, Seoul 06355, Republic of Korea; 3New Drug Development Division, ENCell Corp, Seoul 05854, Republic of Korea

**Keywords:** biological products, dermatitis, atopic, immunoglobulin E, Janus kinase inhibitors

## Abstract

**Background:** Numerous novel medications are being developed to treat moderate to severe atopic dermatitis (AD), notably biologics and Janus kinase inhibitors (JAKis). However, the exact guideline for the first determination of which medication has not been established yet. This study was conducted to identify patients who would show favorable clinical results from using JAKis. **Methods**: Based on the degree of improvement in EASI at 16 weeks, 43 patients were divided into three groups: Group 1 (EASI-90), Group 2 (EASI-75), and Group 3 (EASI-50). **Results**: Compared to Group 1 and Group 2, Group 3 exhibited a significantly higher rate of multiple positive results in MAST (*p*-value = 0.005, *p*-value = 0.004), a greater proportion of individuals with higher IgE levels exceeding 1000 (*p*-value = 0.003, *p*-value = 0.027), and the presence of allergic comorbidities (*p*-value = 0.049, *p*-value = 0.026). However, baseline laboratory test results, such as eosinophil counts, LDH, and so on, showed no significant differences among the three groups. **Conclusions:** JAKis might provide prompt clinical improvement, especially in patients with relatively low serum Ig E levels who do not have multiple allergen positivities and allergic comorbidities.

## 1. Introduction

Atopic dermatitis (AD) is one of the most common and chronically relapsing inflammatory skin diseases that could negatively affect quality of life due to intense pruritus [1,2]. Although the pathophysiology of AD is not completely understood, numerous studies demonstrated that complex interplay of genetic susceptibility, skin barrier dysfunction, immune dysregulation, and environmental factors contribute to the pathobiology of AD. Epidermal barrier impairment, including filaggrin mutations and altered lipid composition, facilitates allergen and microbial penetration, triggering immune activation and systemic sensitization. Type 2 cytokines, such as interleukin (IL)-4, IL-13, and IL-31, play central roles in promoting skin barrier disruption, suppressing antimicrobial peptides, and inducing pruritus. In addition, type 17 and type 22 cytokines, neuroimmune interactions, and microbial dysbiosis contribute to the heterogeneous clinical manifestations of AD. Recent studies have also suggested that dysregulation of the aryl hydrocarbon receptor pathway may further modulate skin barrier function and immune responses in AD [1,3].

Numerous novel medications, including biologics (dupilumab and tralokinumab) and Janus kinase (JAK) inhibitors (upadacitinib, abrocitinib, and baricitinib), have been developed to treat moderate to severe AD refractory to topical therapy [4,5]. However, there is no universal guideline for selecting the best option for each patient. Therefore, patients and physicians should engage in shared decision-making, considering many factors such as efficacy, safety, mode of delivery, cost, speed of onset, and so on [4,5].

Dupilumab and tralokinumab are fully human monoclonal antibodies that can inhibit activities of interleukin (IL)-4/-13 and IL-13, respectively [6,7]. They are administered by subcutaneous injection. In contrast, upadacitinib, abrocitinib, and baricitinib are oral agents that competitively antagonize intracellular Janus kinase/signal transducer and activator of transcription (JAK/STAT) pathway and downward cytokine signaling [8,9,10]. According to several research studies, each of these new targeted immunomodulatory agents has shown high efficacy and safety [6,7,8,9,10,11]. Absolute or relative contraindications and side effects that occur during treatment can lead to selection of another class of drugs. However, which drug should be chosen as the primary treatment for specific patient groups remains to be determined.

High costs and strict insurance regulations in Korea make switching between two medications complex. However, to date, there are no validated predictive markers or stratification strategies to guide the selection between JAK inhibitors and biologics in clinical practice. The development of predictive tools is crucial to enable more personalized and effective systemic treatment choices in moderate to severe AD. This research was designed to identify patients who would obtain favorable clinical results from using JAK inhibitors among patients with moderate to severe AD. It also aimed to suggest possible guidelines for using JAK inhibitors as primary treatment options for specific patient groups.

## 2. Materials and Methods

### 2.1. Patients and Inclusion Criteria

This was a single-center retrospective observational study approved by the Institutional Review Board (IRB) of Samsung Medical Center (SMC), Sungkyunkwan University School of Medicine in Seoul, Korea (IRB approval number: SMC 2024-12-019). We collected data from adult AD patients aged over 18 years who visited the dermatology center between January 2021 and November 2024 and received a minimum of 16 weeks of JAK inhibitor treatment. Patients who irregularly received JAK inhibitors and those whose clinical evaluation data (eczema area and severity index (EASI)) at baseline and week 16 were unavailable were excluded from this study.

### 2.2. Patient Categorization

All patients in the analysis demonstrated clinical improvement. To assess and compare outcomes, patients were categorized into three groups based on the degree of improvement according to their EASI scores. Group 1 (EASI-90) included patients with 90% or more improvement in their EASI scores. Group 2 (EASI-75) consisted of those with improvements of 75% or more but less than 90%, while Group 3 (EASI-50) comprised patients with improvements ranging from 50% to less than 75%. Dermatologic specialists carefully assessed all EASI scores at distinct timepoints: baseline (T0) and week 16 (T1).

### 2.3. Demographic and Laboratory Data Collection

For each group, demographic data including age, sex, height, weight, AD onset time, and allergic comorbidities were collected, along with laboratory data. Peak pruritus numerical rating scale (PP-NRS) developed to assess itch intensity in AD was also collected. Laboratory data included immunoglobulin E (IgE), eosinophil count, lactate dehydrogenase (LDH), eosinophil cationic protein (ECP), uric acid, and results of multiple allergen simultaneous test (MAST) to assess allergen-specific IgE levels. Serum IgE and eosinophil count could indicate type 2 inflammation, with ECP serving as a marker of eosinophil activity. Both LDH and uric acid could serve as indicators of cellular turnover and systemic inflammation.

### 2.4. Classification Criteria for Predictive Value

Allergen-specific IgE was classified as positive or negative based on MAST results. At the same time, baseline IgE levels were categorized as high or low based on achieving the lowest *p*-value in group comparisons. A cut-off value of 1000 of IgE was determined through this statistical analysis. However, no significant cut-off values were identified for other biomarkers such as eosinophil count or LDH. Allergic comorbidities were grouped based on the presence or absence of conditions such as asthma, allergic rhinitis, allergic conjunctivitis, and food allergies.

### 2.5. Statistical Analyses

Data were analyzed with R 4.3.1 (R Core Team 2023, Vienna, Austria). Analysis of variance (ANOVA) was used for continuous variables, while Chi-square test and Fisher–Freeman–Halton test were employed for categorical variables to compare demographics and predictive factors at each timepoint. Post hoc analysis was performed when significant differences were observed among three groups. A *p*-value of less than 0.05 was considered statistically significant.

Cut-off values were determined for laboratory tests, including IgE, eosinophil count, LDH, ECP, and uric acid. Each value was dichotomized, assigning 1 for values above the threshold and 0 for those below the threshold. Chi-square tests were then conducted to assess differences in the proportion of cases classified as 1 across groups. The cut-off point was selected as the value corresponding to the lowest *p*-value.

## 3. Results

### 3.1. Demographics

A total of 43 patients were assessed in this study. Among them, 11, 18, and 14 patients were classified into Group 1, Group 2, and Group 3, respectively (Table 1). Of the 43 patients, 30 (69.77%) were males and 13 (30.23%) were females. Patients’ ages ranged from 19 to 76 years, with a mean age of 31.5 years. When the onset of AD was investigated, it was revealed that 23, 8, and 12 patients developed the condition during childhood, during adolescence, and during adulthood, respectively. Additionally, there were no significant differences in gender, age, AD onset timing, weight, height, or body mass index (BMI) among the three groups.

The average baseline EASI score was 21.25 overall, 26.36 for Group 1, 21.72 for Group 2, and 16.65 for Group 3. Although their differences were not statistically significant, there was a trend toward lower severity in Group 3.

Three types of JAK inhibitors were used in this study: upadacitinib, abrocitinib, and baricitinib. Among these, upadacitinib was predominantly prescribed, with 86.05% of patients receiving it, while abrocitinib and baricitinib were each used by 6.98% of patients.

### 3.2. Laboratory Test and PP-NRS Results: Baseline vs. Week 16

Baseline and week 16 measurements of IgE, eosinophil count, ECP, LDH, and uric acid were collected (Table 2). Since patients visited the hospital at week 16 for insurance evaluation and EASI score assessment, laboratory tests were also performed at that time. Laboratory results showed no significant differences among the three groups. Notably, no group showed a reduction exceeding 10% from baseline at week 16 for IgE levels. In particular, Group 3 demonstrated an increase in IgE level at week 16 compared to baseline. However, no significant differences were found when comparing differences (Baseline–Week 16) in laboratory results across the three groups at each timepoint. The PP-NRS showed a remarkable improvement in all three groups within 16 weeks, with the majority of patients achieving a mild score (0–3) by week 16 (Table 2).

### 3.3. Predictive Factors

Several predictive factors were compared among the three groups (Table 3). The EASI score showed no significant difference among the three groups at baseline, whereas a significant difference was observed at week 16. At baseline, Group 3 showed a trend toward lower severity. However, at week 16, Group 1 exhibited the lowest EASI score. This outcome was expected as groups were assigned based on the degree of EASI improvement. Additionally, the proportion of total EASI score attributed to head and neck regions showed no significant difference among the three groups.

For allergen-specific IgE positivity determined by MAST, Group 1 had two positive and eight negative results, Group 2 had four positive and twelve negative results, and Group 3 had eight positive and one negative result, demonstrating a significant difference in distribution among the three groups.

Baseline IgE levels categorized with a cut-off value of 1000 also showed significant differences. In Group 1, five and six patients were classified as high and low, respectively. In Group 2, twelve and six were classified as high and low, respectively. In Group 3, all thirteen patients were classified as high.

Regarding allergic comorbidities including asthma, allergic rhinitis, allergic conjunctivitis, and food allergy, Group 1 had two patients with comorbidities and nine without, Group 2 had three with comorbidities and fifteen without, and Group 3 had eight with comorbidities and six without, showing a significant difference across the three groups.

### 3.4. Predictive Factors: Subgroup Analysis

Post hoc multiple comparisons were conducted to identify which groups showed significant differences in allergen-specific IgE, baseline IgE, and allergic comorbidities (Figure 1). Group 1 and Group 2 showed no significant differences for any of these factors. In contrast, Group 1 and Group 3 showed significant differences in allergen-specific IgE, baseline IgE, and allergic comorbidities, with *p*-values of 0.005, 0.003, and 0.049, respectively. Similarly, Group 2 and Group 3 showed significant differences in allergen-specific IgE, baseline IgE, and allergic comorbidities, with *p*-values of 0.004, 0.027, and 0.026, respectively. Multivariable regression analyses adjusting for potential confounders were performed. The results were consistent with the univariate analyses, with no significant alterations in the direction or magnitude of associations.

## 4. Discussion

AD is a chronic inflammatory skin disease that places a significant disease burden on individual patients [12]. It requires lifelong treatment. In cases of frequent flare-ups or poor control, it can negatively affect patients’ overall health, including itching, sleep disturbances, and depression [12,13]. Additionally, the high cost of immunomodulatory therapies such as biologics and JAK inhibitors can be burdensome, making it important for patients to seek support from the insurance system to help with treatment costs [14].

For insurance coverage, the EASI score at week 16 after starting an immunomodulatory agent must show an improvement of at least 75% compared to the baseline EASI score [15]. In our study, we compared Groups 1/2 (EASI improvement ≥ 75%) and Group 3 (EASI improvement < 75%) and found that initial treatment starting with JAK inhibitors was not a favorable option for patients with multiple allergen sensitivities, baseline IgE levels > 1000 IU/mL, and allergic comorbidities. Even among patients with head and neck involvement, the effectiveness of JAK inhibitors demonstrated limited efficacy in those with high serum IgE levels at baseline. Furthermore, JAK inhibitors did not result in a substantial reduction in IgE levels over 16 weeks, suggesting that serum IgE may not serve as a reliable predictive biomarker for disease monitoring. In contrast, dupilumab has been shown to effectively reduce IgE levels over time, particularly in patients with well-controlled disease [3]. A recent study demonstrated that patients who underwent dupilumab treatment for at least 40 weeks maintained lower serum IgE levels even after dose tapering, supporting the notion that dupilumab may have a direct impact on IgE regulation [3]. This distinction in IgE modulation between JAK inhibitors and dupilumab may be an important consideration when selecting treatment strategies for patients with AD.

Several previous studies have provided insights into drug selection strategies for biologics and JAK inhibitors. Kamata et al. [4] highlighted that biologics may be preferred for patients at risk for malignancy, cardiovascular disease (CVD), deep vein thrombosis, pulmonary embolism, or gastrointestinal perforation. They also recommended biologics for elderly patients over 50 years, particularly those over 65 years, and individuals with a history of herpes zoster, recurrent skin infections (including eczema herpeticum), or comorbidities such as asthma, eosinophilic esophagitis, and chronic rhinosinusitis with nasal polyps. In contrast, JAK inhibitors have been suggested for patients with severe pruritus requiring rapid onset of relief, those who fear needles or prefer oral medication, and individuals at high risk of conjunctivitis during dupilumab treatment. JAK inhibitors were also indicated for facial redness, alopecia areata, or arthritis/enthesitis associated with biologics. Kim et al. [5] proposed that biologics might be safer for higher-risk and elderly patients, whereas higher doses of abrocitinib and upadacitinib demonstrate greater efficacy than biologics. Similarly, Calabrese et al. [16] emphasized that biologics are suitable for patients with CVD, children and elderly populations, those who have extrinsic AD, European American ethnicity, asthma, recurrent varicella-zoster virus infections, oncological comorbidities, and acne. In contrast, JAK inhibitors were recommended for patients with ocular surface disease, severe AD with intense itching, intrinsic AD, Asian or African American ethnicity, head and neck AD, vitiligo, inflammatory bowel disease, and alopecia areata. Additionally, this study provides a possible guideline for identifying which laboratory findings and patient histories should be considered before initiating immunomodulatory agents (Table 4).

This study contributes to the evidence highlighting the need for immunologically personalized approaches in AD management. AD is a complex and immunologically heterogeneous systemic disease that requires an understanding of its diverse inflammatory pathways for effective treatment. The pathophysiology of AD is primarily driven by type 2 inflammation, which plays a crucial role in disease progression [17]. In the acute phase of AD [1,18], the impaired skin barrier affected by genetic and environmental conditions allows antigens to penetrate and interact with antigen-presenting cells such as Langerhans cells and dermal dendritic cells. These cells can activate naïve T cells, which can differentiate into T-helper 2 (TH2) cells under the influence of IL-4. TH2 cells play a central role in the initial inflammatory process by releasing cytokines such as IL-4 and IL-13. These cytokines can exacerbate skin barrier dysfunction and stimulate B cells to undergo class switching, leading to IgE production and activation of mast cells and eosinophils. Additionally, IL-31, another TH2 cytokine, contributes to the characteristic pruritus seen in AD [19]. T-helper 22 (TH22) cells further contribute to the disease pathology by producing IL-22, which can synergize with IL-17 secreted by T-helper 17 (TH17) cells to promote epidermal hyperplasia and skin thickening [20]. In the chronic phase [21], the immune response diversifies, with involvement of T-helper 1 (TH1), TH17, and TH22 cells intensifying effects of TH2 cytokines. For example, TH1 cells can produce interferon (IFN)-γ that can amplify inflammatory conditions and perpetuate the disease cycle [21].

Not all AD cases exhibit TH2-dominant inflammation [22]. Instead, AD encompasses a spectrum of immune profiles, with some patients demonstrating significant contributions from TH1, TH17, or TH22 pathways. Extrinsic and intrinsic AD represent two distinct subtypes of the disease. They can be characterized by differences in immunological profiles, allergen sensitivities, and clinical presentations [23,24]. Extrinsic AD accounts for approximately 80% of total AD cases. It is closely associated with allergic comorbidities such as asthma and allergic rhinitis. Patients with extrinsic AD typically exhibit elevated serum total IgE levels and allergen-specific IgE, with a predominant sensitivity to protein allergens. In contrast, intrinsic AD comprises about 20% of AD cases. It is less commonly linked to allergic comorbidities. Serum total and allergen-specific IgE levels in intrinsic AD patients tend to remain within normal ranges, with sensitization often involving non-protein allergens such as metals and haptens. Immunologically, extrinsic AD is characterized by a dominance of the TH2 pathway, with increased levels of cytokines such as IL-4, IL-13, and IL-5. Conversely, intrinsic AD demonstrates a more complex immune profile involving TH1, TH17, and TH22 pathways in addition to TH2^22^. IFN-γ in particular can suppress IgE production, distinguishing intrinsic AD from the extrinsic subtype that is IgE-dominated.

Biologics such as dupilumab and tralokinumab primarily target type 2 inflammation by inhibiting IL-4, IL-13, or IL-13 alone, making them effective for patients whose AD is predominantly driven by the TH2 pathway [17,25]. However, JAK inhibitors, which block multiple cytokines including TH2, TH1, TH17, and TH22 cytokines by inhibiting the JAK/STAT signaling cascade, might be more suitable for patients with mixed or non-type 2 inflammatory profiles such as those with significant TH1 or TH17 involvement [26,27]. Given these differences, JAK inhibitors can be considered to be effective in the treatment of patients showing complex features of immunologic responses and intrinsic AD.

Our study revealed that the PP-NRS at week 16 improved to mild levels in almost all cases regardless of the group. This finding suggests that JAK inhibitors can effectively alleviate itch rapidly. Itch in the acute phase of AD is associated with several histaminergic pathways [28]. Among TH2 cytokines, IL-4, IL-13, IL-31, and thymic stromal lymphopoietin (TSLP) secreted by keratinocytes are known to play critical roles. Scratching can induce keratinocytes to release TSLP, perpetuating the itch–scratch cycle. TH17 cytokines such as IL-17 and IL-23 are also implicated in pruritus. As the JAK/STAT pathway is involved in the pathogenesis of itch in AD, JAK inhibitors may be prioritized for patients with severe itch.

Our findings are also consistent with other recent real-world studies that have demonstrated the effectiveness and the safety of JAK inhibitors. These studies similarly reported substantial improvements in disease severity scores (e.g., EASI, PP-NRS) within the first 12 to 16 weeks of treatment [29,30,31]. In addition, no serious adverse events were observed during the 16-week treatment period of our study. Mild adverse events included headache, gastrointestinal discomfort, acne, transient elevation of liver enzymes, and transient dyslipidemia, which are in line with known class effects of JAK inhibitors [3,30,32]. Previous real-world studies have similarly reported that while laboratory abnormalities such as transient increases in lipid parameters (LDL, HDL cholesterol) and liver enzymes are common, these changes are typically mild, non-progressive, and manageable with routine monitoring [30,32,33]. Additionally, the incidence of serious infections and thromboembolic events remains low in short-term real-world analyses [32]. These findings reinforce the importance of individualized risk assessment and monitoring when initiating JAK inhibitor therapy in AD patients, particularly in those with pre-existing metabolic or cardiovascular risk factors.

In conclusion, from a clinical perspective, these results may offer preliminary insights to guide treatment selection. For example, in patients with markedly elevated serum IgE levels and multiple allergic comorbidities, biologic agents targeting type 2 cytokines (such as dupilumab) may be favored as first-line systemic therapy. Conversely, JAK inhibitors could represent an effective option for patients with lower IgE levels, fewer allergic features, or those with significant pruritus requiring rapid symptomatic relief. This study has several limitations. It is a single-center, retrospective analysis with a relatively small sample size, and the patient cohort was not prospectively stratified by intrinsic or extrinsic subtypes. The IgE cut-off used for subgroup analysis was derived post hoc and requires validation in larger prospective cohorts. In addition, long-term outcomes beyond 16 weeks were not evaluated. Therefore, larger prospective multi-center studies are necessary to validate these findings and to refine patient selection criteria for the optimal use of systemic treatments in AD.

## Figures and Tables

**Figure 1 jcm-14-04312-f001:**
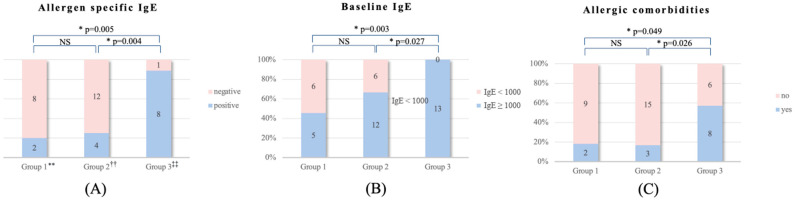
Subgroup analysis of each predictive factor for three groups. (**A**) Allergen-specific immunoglobulin E measured by multiple allergen simultaneous test, (**B**) baseline immunoglobulin E divided by cut-off value 1000, and (**C**) allergic comorbidities including asthma, allergic rhinitis, allergic conjunctivitis, and food allergy. **, Patients who improved by 90% or more in their EASI scores; ^††^, patients who improved by 75% or more but less than 90% in their EASI scores; and ^‡‡^, patients who improved by 50% or more but less than 75% in their EASI scores.

**Table 1 jcm-14-04312-t001:** Demographics of patients.

	Overall (n = 43)	Group 1 * (n = 11)	Group 2 † (n = 18)	Group 3 ‡ (n = 14)	*p*-Value
Gender, male	30 (69.77)	9 (81.82)	13 (72.22)	8 (57.15)	0.393
Age, years	31.51 (12.57)	32.27 (13.31)	26.61 (6.43)	37.21 (15.84)	0.155
AD onset					0.646
Childhood	23 (53.49)	7 (63.64)	10 (55.56)	6 (42.86)	
Adolescence	8 (18.60)	2 (18.18)	4 (22.22)	2 (14.26)	
Adulthood	12 (27.91)	2 (18.18)	4 (22.22)	6 (42.46)	
Weight (kg)	72.76 (15.33)	74.41 (11.19)	72.93 (15.73)	71.43 (18.14)	0.908
Height (cm)	168.57 (9.33)	172.56 (6.82)	167.74 (7.52)	166.77 (12.15)	0.334
BMI (kg/m^2^)	25.72 (4.73)	24.95 (3.05)	26.40 (5.64)	25.47 (4.79)	0.909
Baseline EASI	21.25 (10.33)	26.36 (6.89)	21.72 (9.62)	16.65 (11.92)	0.06
JAK inhibitors					
Upadacitinib	37 (86.05)	11 (100)	16 (88.89)	10 (71.43)	
Abrocitinib	3 (6.98)	0 (0)	2 (11.11)	1 (7.14)	
Baricitinib	3 (6.98)	0 (0)	0 (0)	3 (21.43)	

*, Patients with a 90% or greater improvement in their EASI scores; †, patients with an improvement of at least 75% but less than 90% in their EASI scores; and ‡, patients with an improvement of at least 50% but less than 75% in their EASI scores. Abbreviations: AD, atopic dermatitis; BMI, body mass index; EASI, eczema area and severity index; JAK, Janus kinase.

**Table 2 jcm-14-04312-t002:** Biomarker results for each group at baseline and week 16.

		Group 1 * (n = 11)	Group 2 † (n = 18)	Group 3 ‡ (n = 14)	*p*-Value
IgE (IU/mL)	Baseline	2626.14 (2232.47)	2691.35 (1907.44)	2953.09 (2101.06)	0.925
	Week 16	2700.13 (2539.43)	2494.44 (2122.14)	4190.00 (1682.50)	0.38
	Δ (Baseline–Week 16)	43.88 (115.55)	−264.97 (849.78)	−188.50 (811.76)	0.749
Eosinophil (/μL)	Baseline	507.09 (487.29)	860.20 (659.13)	456.14 (257.53)	0.214
	Week 16	248.50 (144.86)	424.38 (514.04)	260.00 (103.15)	0.565
	Δ (Baseline–Week 16)	−462.83 (612.97)	−501.38 (695.63)	−58.33 (141.76)	0.32
ECP (μg/L)	Baseline	108.05 (131.45)	55.12 (26.63)	36.37 (20.22)	0.439
	Week 16	38.00 (N/A)	18.30 (6.51)	8.51 (N/A)	N/A
	Δ (Baseline–Week 16)	−163.00 (N/A)	−16.25 (14.07)	−4.99 (N/A)	N/A
LDH (IU/L)	Baseline	314.17 (153.80)	258.86 (70.49)	199.00 (32.16)	0.097
	Week 16	261.00 (44.50)	257.33 (83.10)	190.56 (108.37)	0.428
	Δ (Baseline–Week 16)	−124.67 (149.18)	−15.33 (129.16)	61.50 (24.93)	0.142
Uric acid (mg/dL)	Baseline	5.86 (0.98)	6.78 (1.44)	5.83 (1.83)	0.282
	Week 16	5.85 (1.33)	5.90 (0.96)	6.24 (0.44)	0.786
	Δ (Baseline–Week 16)	−0.25 (0.72)	−0.87 (1.26)	0.16 (0.98)	0.303
PP-NRS	Baseline	8.125 (1.46)	7.625 (1.51)	6.429 (2.07)	
	Week 16	1.200 (1.23)	1.643 (1.28)	2.000 (1.49)	

*, Patients with a 90% or greater improvement in their EASI scores; †, patients with an improvement of at least 75% but less than 90% in their EASI scores; and ‡, patients with an improvement of at least 50% but less than 75% in their EASI scores. Abbreviations: ECP, eosinophil cationic protein; IgE, immunoglobulin E; LDH, lactate dehydrogenase; N/A, not available; and PP-NRS: peak pruritus numerical rating scale.

**Table 3 jcm-14-04312-t003:** Comparison of Clinical characteristics and laboratory results among three groups.

	Group 1 * (n = 11)	Group 2 † (n = 18)	Group 3 ‡ (n = 14)	*p*-Value
Baseline EASI, mean (SD)	26.35 (6.89)	21.72 (9.62)	16.65 (11.92)	0.06
Week 16 EASI, mean (SD)	1.23 (0.76)	3.45 (1.68)	8.86 (8.65)	0.001
Allergen-specific IgE (MAST results)				0.002
Positive	2	4	8	
Negative	8	12	1	
Baseline IgE				0.011
IgE ≥ 1000	5	12	13	
IgE < 1000	6	6	0	
Allergic comorbidity				0.028
Yes	2	3	8	
No	9	15	6	
Head and neck/EASI ratio (%) median [IQR]	0.14 [0.10, 0.17]	0.11 [0.07, 0.16]	0.14 [0.05, 0.25]	0.79

*, Patients with a 90% or greater improvement in their EASI scores; †, patients with an improvement of at least 75% but less than 90% in their EASI scores; and ‡, patients with an improvement of at least 50% but less than 75% in their EASI scores. Abbreviations: EASI, eczema area and severity index; IgE, immunoglobulin E; IQR, interquartile range; and MAST, multiple allergen simultaneous test.

**Table 4 jcm-14-04312-t004:** Summary of Clinical Recommendations for Biologics vs. JAK Inhibitors Based on Patient Profiles in AD.

Study	Biologics	Janus Kinase Inhibitor
Kamata et al. [4] 2023	Risk of malignancy, CVD, DVT, PE, GI perforation Elderly patients (over 50 years, especially over 65 years) History of herpes zoster Repeated skin infections including eczema herpeticum Comorbidities: asthma, eosinophilic esophagitis, and/or chronic rhinosinusitis with nasal polyps	Severe pruritus and/or who wish a rapid onset Fear of needles (trypanophobia) or who prefers oral medicine over injection High risk of developing conjunctivitis during dupilumab treatment (history of conjunctivitis, etc.) Facial redness, alopecia areata, or arthritis/enthesitis caused by biologics
Kim et al. [5] 2024	Higher-risk and elderly patients (considered safer than JAK inhibitors)	Higher doses of abrocitinib and upadacitinib are more effective than biologics
Calabrese et al. [16] 2024	CVD comorbidities Children and elderly Extrinsic AD European American Asthma Recurrent VZV infections Oncological comorbidities Acne	Ocular surface disease Severe AD and itch Intrinsic AD Asian and African American Head and neck AD Vitiligo Inflammatory bowel disease Alopecia areata
Present study	Extrinsic AD	Intrinsic AD Multiple allergen positivities Relatively low baseline IgE levels below 1000 Without allergic comorbidities (asthma, allergic rhinitis, allergic conjunctivitis, and food allergy)

Abbreviations: AD, atopic dermatitis; CVD, cardiovascular disease; DVT, deep vein thrombosis; PE, pulmonary embolism; GI, gastrointestinal; and VZV, varicella-zoster virus.

## Data Availability

The data that support the findings of this study are available from the corresponding author upon reasonable request.

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
