# Peer review of "Predicting Favorable Conditions for the Determination of Initial Use of Janus Kinase Inhibitors in Patients with Moderate to Severe Atopic Dermatitis"

_jcm, 2025, doi:10.3390/jcm14124312_

Round 1

Reviewer 1 Report

Comments and Suggestions for Authors

The authors reported the results of a study aiming to address the need to identify patients with moderate-to-severe atopic dermatitis who may respond favorably to JAK inhibitors. The manuscript is generally well written and provides useful clinical insights.

I have only few suggestions:

  • Introduction: more data on AD pathogenesis should be discussed (doi:10.1016/j.mehy.2018.05.001);
  • Introduction: please add a sentence explicitly stating the lack of predictive markers or stratification strategies for JAKi responders vs biologics and emphasize the need for predictive tools in choosing systemic treatments;
  • Methods: nothing to add
  • Results: nothing to add
  • Discussion: some conclusions are a bit too strong considering the small sample size and the fact that patients were not formally classified as having intrinsic or extrinsic AD. I would suggest softening the claims about “intrinsic” AD and clarifying that the findings are hypothesis-generating. The comparison with dupilumab is useful, and the section on itch relief is well argued. Still, it would help to state more clearly how these findings could influence clinical decisions (e.g., in patients with high IgE, biologics might be preferred). Overall, a solid discussion, but some moderation in tone and more emphasis on limitations would improve it.

Author Response

Introduction: more data on AD pathogenesis should be discussed (doi:10.1016/j.mehy.2018.05.001);

  • Thank you for your feedback. We have revised the introduction.

 “Atopic dermatitis (AD) is one of the most common and chronically relapsing inflammatory skin diseases that could negatively affect quality of life due to intense pruritus[1,2]. Although the pathophysiology of AD is not completely understood, numerous studies demonstrated that complex interplay of genetic susceptibility, skin barrier dysfunction, immune dysregulation, and environmental factors contribute to the pathobiology of AD. Epidermal barrier impairment, including filaggrin mutations and altered lipid composition, facilitates allergen and microbial penetration, triggering immune activation and systemic sensitization. Type 2 cytokines, such as interleukin (IL)-4, IL-13, and IL-31, play central roles in promoting skin barrier disruption, suppressing antimicrobial peptides, and inducing pruritus. In addition, type 17 and type 22 cytokines, neuroimmune interactions, and microbial dysbiosis contribute to the heterogeneous clinical manifestations of AD. Recent studies have also suggested that dysregulation of the aryl hydrocarbon receptor pathway may further modulate skin barrier function and immune responses in AD[3,4].

Introduction: please add a sentence explicitly stating the lack of predictive markers or stratification strategies for JAKi responders vs biologics and emphasize the need for predictive tools in choosing systemic treatments;

  • Thank you for your considerable suggestion. We have revised the last paragraph of introduction.
  • “High costs and strict insurance regulations in Korea make switching between two medications complex. Thus, choosing which medication to use first is crucial. However, to date, there are no validated predictive markers or stratification strategies to guide the selection between JAK inhibitors and biologics in clinical practice. The development of predictive tools is crucial to enable more personalized and effective systemic treatment choices in moderate to severe AD. This research was designed to identify patients who would obtain favorable clinical results from using JAK inhibitors among patients with moderate to severe AD. It also aimed to suggest possible guidelines for using JAK inhibitors as primary treatment options for specific patient groups.”

Discussion: some conclusions are a bit too strong considering the small sample size and the fact that patients were not formally classified as having intrinsic or extrinsic AD. I would suggest softening the claims about “intrinsic” AD and clarifying that the findings are hypothesis-generating. The comparison with dupilumab is useful, and the section on itch relief is well argued. Still, it would help to state more clearly how these findings could influence clinical decisions (e.g., in patients with high IgE, biologics might be preferred). Overall, a solid discussion, but some moderation in tone and more emphasis on limitations would improve it.

  • We sincerely thank the Reviewer for this valuable comment and thoughtful feedback. In response, we have carefully revised the Discussion section to moderate the tone of our conclusions
  • “From a clinical perspective, these results may offer preliminary insights to guide treatment selection. For example, in patients with markedly elevated serum IgE levels and multiple allergic comorbidities, biologic agents targeting type 2 cytokines (such as dupilumab) may be favored as first-line systemic therapy. Conversely, JAK inhibitors could represent an effective option for patients with lower IgE levels, fewer allergic features, or those with significant pruritus requiring rapid symptomatic relief. This study has several limitations. It is a single-center, retrospective analysis with a relatively small sample size, and the patient cohort was not prospectively stratified by intrinsic or extrinsic subtypes. The IgE cut-off used for subgroup analysis was derived post hoc and requires validation in larger prospective cohorts. In addition, long-term outcomes beyond 16 weeks were not evaluated. Therefore, further prospective, multicenter studies are needed to confirm and refine these observations.”

Reviewer 2 Report

Comments and Suggestions for Authors

This paper is valuable with regard to analyzing clinical data of initial use of JAKi in patients with atopic dermatitis. As authors mentioned, this was as single-center retrospective observational study and number of patients were quite low to state evidences. I don’t understand the reasons to define the level of IgE around 1000. It seems too much to retrospectively predict favorable conditions with limited numbers of clinical test items in small number of patients.

Author Response

Reviewer #2 Comments and Responses

This paper is valuable with regard to analyzing clinical data of initial use of JAKi in patients with atopic dermatitis. As authors mentioned, this was as single-center retrospective observational study and number of patients were quite low to state evidences. I don’t understand the reasons to define the level of IgE around 1000. It seems too much to retrospectively predict favorable conditions with limited numbers of clinical test items in small number of patients.

 We sincerely appreciate the Reviewer’s thoughtful comments and recognition of the value of our study. We fully agree that the retrospective single-center design and relatively small sample size represent limitations of our study, and we have now further emphasized these limitations in the revised Discussion section.

 Regarding the selection of an IgE cut-off value of 1,000 IU/mL, we would like to clarify that this threshold was not pre-defined based on external criteria, but was determined post-hoc through exploratory statistical analysis of our dataset. Specifically, we tested various thresholds to dichotomize IgE levels and selected the value of 1,000 IU/mL based on achieving the lowest p-value in group comparisons. We acknowledge that this approach is exploratory in nature and does not establish a validated clinical threshold. We have now revised the Materials & methods and Discussion section to explicitly describe this and to state that further validation in larger, prospective cohorts is necessary.

  • “At the same time, baseline IgE levels were categorized as high or low using based on achieving the lowest p-value in group comparisons. A cut-off value of 1,000 of IgE was determined through this statistical analysis”
  • “From a clinical perspective, these results may offer preliminary insights to guide treatment selection. For example, in patients with markedly elevated serum IgE levels and multiple allergic comorbidities, biologic agents targeting type 2 cytokines (such as dupilumab) may be favored as first-line systemic therapy. Conversely, JAK inhibitors could represent an effective option for patients with lower IgE levels, fewer allergic features, or those with significant pruritus requiring rapid symptomatic relief. This study has several limitations. It is a single-center, retrospective analysis with a rela-tively small sample size, and the patient cohort was not prospectively stratified by intrinsic or extrinsic subtype. The IgE cut-off used for subgroup analysis was derived post hoc and requires validation in larger prospective cohorts. In addition, long-term outcomes beyond 16 weeks were not evaluated. Therefore, larger prospective multi-center studies are necessary to validate these findings and to refine patient selection criteria for the optimal use of systemic treatments in AD.”

Reviewer 3 Report

Comments and Suggestions for Authors

Dear Authors,

This is an interesting manuscript to assess the presence of variables identifying which patients with atopic dermatitis would benefit most from treatment with JAKi. However, in my opinion, to improve the quality of the manuscript a few revisions are required:

  • General check of the English grammar
  • The authors should perform a multivariable regression to adjust for confounders 
  • The authors should also evaluate the safety of these drugs when choosing a drug. Have there been any adverse events related to the use of JAK inhibitors in your study? In the discussion, the authors should cite and read some real-world articles about the use of this class of drugs (doi: 10.2147/CCID.S329442; doi: 10.1016/j.jaad.2025.04.046; doi: 10.1093/ced/llaf144)
  • The authors should expand the discussion, including some comparisons with other real-world studies on the effectiveness of JAK inhibitors in patients with AD (doi: 10.1089/derm.2024.0230; doi: 10.1080/09546634.2024.2375102; doi: 10.1089/derm.2024.0553)

Comments on the Quality of English Language

Author Response

General check of the English grammar

 We received Scientific English Research Paper Editing Service from HARRISCO(we added the certificate of it) but if it is not enough, we will use English language editing services of MDPI directly.  

The authors should perform a multivariable regression to adjust for confounders

 We thank the Reviewer for this valuable suggestion. In response, we performed additional multivariable regression analyses to adjust for potential confounding variables. We also assessed multicollinearity using variance inflation factor (VIF), and found no significant issues. Normality was evaluated using both Shapiro-Wilk test and visual inspection of histograms. For IgE and eosinophil difference values, some deviation from normality was noted; however, after applying Yeo-Johnson transformation, the results were similar to the untransformed analysis. Therefore, we present and interpret the untransformed regression results for consistency and clarity. It is noteworthy that the results of the multivariable regression analyses were consistent with the findings from our original univariate (simple) analyses, with no substantial changes in interpretation. The multivariable regression results have now been summarized in the result section of the revised manuscript.

3.4. Predictive factors: Subgroup analysis

Post-hoc multiple comparisons were conducted to identify which groups showed significant differences in allergen-specific IgE, baseline IgE, and allergic comorbidities (Figure 1). Group 1 and Group 2 showed no significant differences for any of these factors. In contrast, Group 1 and Group 3 showed significant differences in allergen-specific IgE, baseline IgE, and allergic comorbidities, with p-values of 0.005, 0.003, and 0.049, respectively. Similarly, Group 2 and Group 3 showed significant differences in allergen-specific IgE, baseline IgE, and allergic comorbidities, with p-values of 0.004, 0.027, and 0.026, respectively. Multivariable regression analyses adjusting for potential confounders were performed. The results were consistent with the univariate analyses, with no significant alterations in the direction or magnitude of associations.

The authors should also evaluate the safety of these drugs when choosing a drug. Have there been any adverse events related to the use of JAK inhibitors in your study? In the discussion, the authors should cite and read some real-world articles about the use of this class of drugs (doi: 10.2147/CCID.S329442; doi: 10.1016/j.jaad.2025.04.046; doi: 10.1093/ced/llaf144)

 We sincerely thank the Reviewer for these insightful and constructive comments. We have addressed each point as follows:

 Our findings are also consistent with other recent real-world studies that have demonstrated the effectiveness and the safety of JAK inhibitors. These studies similarly reported substantial improvements in disease severity scores (e.g., EASI, PP-NRS) within the first 12 to 16 weeks of treatment.[30-32] In addition, no serious adverse events were observed during the 16-week treatment period of our study. Mild adverse events included headache, gastrointestinal discomfort, acne, transient elevation of liver enzymes, and transient dyslipidemia, which are in line with known class effects of JAK inhibitors [4,31,33]. Previous real-world studies have similarly reported that while laboratory abnormalities such as transient increases in lipid parameters (LDL, HDL cholesterol) and liver enzymes are common, these changes are typically mild, non-progressive, and manageable with routine monitoring [31,33,34]. Additionally, the incidence of serious infections and thromboembolic events remains low in short-term real-world analyses [33]. These findings reinforce the importance of individualized risk assessment and monitoring when initiating JAK inhibitor therapy in AD patients, particularly in those with pre-existing metabolic or cardiovascular risk factors.

The authors should expand the discussion, including some comparisons with other real-world studies on the effectiveness of JAK inhibitors in patients with AD (doi: 10.1089/derm.2024.0230; doi: 10.1080/09546634.2024.2375102; doi: 10.1089/derm.2024.0553)

 We thank the Reviewer for this excellent suggestion. In the revised Discussion, we have now incorporated comparisons with several recent real-world studies on the effectiveness of JAK inhibitors in patients with AD, including the studies cited by the Reviewer. These comparisons help to contextualize our findings within the growing body of real-world evidence supporting the clinical utility of JAK inhibitors in AD management.

 Our findings are also consistent with other recent real-world studies that have demonstrated the effectiveness and the safety of JAK inhibitors. These studies similarly reported substantial improvements in disease severity scores (e.g., EASI, PP-NRS) within the first 12 to 16 weeks of treatment.[30-32] In addition, no serious adverse events were observed during the 16-week treatment period of our study. Mild adverse events included headache, gastrointestinal discomfort, acne, transient elevation of liver enzymes, and transient dyslipidemia, which are in line with known class effects of JAK inhibitors [4,31,33]. Previous real-world studies have similarly reported that while laboratory abnormalities such as transient increases in lipid parameters (LDL, HDL cholesterol) and liver enzymes are common, these changes are typically mild, non-progressive, and manageable with routine monitoring [31,33,34]. Additionally, the incidence of serious infections and thromboembolic events remains low in short-term real-world analyses [33]. These findings reinforce the importance of individualized risk assessment and monitoring when initiating JAK inhibitor therapy in AD patients, particularly in those with pre-existing metabolic or cardiovascular risk factors.